# Inactivation of Airborne Avian Pathogenic *E. coli* (APEC) via Application of a Novel High-Pressure Spraying System

**DOI:** 10.3390/microorganisms10112201

**Published:** 2022-11-07

**Authors:** Dongryeoul Bae, Kwang-Young Song, Donah Mary Macoy, Min Gab Kim, Chul-Kyu Lee, Yu-Seong Kim

**Affiliations:** 1College of Veterinary Medicine, Konkuk University, Seoul 05029, Korea; 2Research Institute of Pharmaceutical Science, College of Pharmacy, Gyeongsang National University, Jinju 52828, Korea; 3Division of Research and Development, TracoWorld Ltd., Gwangmyeong-si 14348, Korea

**Keywords:** chick, avian pathogenic *E. coli*, colibacillosis, spray, space disinfection

## Abstract

Infectious diseases of livestock caused by novel pathogenic viruses and bacteria are a major threat to global animal health and welfare and their effective control is crucial for agronomic health and for securing global food supply. It has been widely recognized that the transmission of infectious agents can occur between people and/or animals in indoor spaces. Therefore, infection control practices are critical to reduce the transmission of the airborne pathogens. ViKiller^®^-high-pressure sprayer and Deger^®^-disinfectant are newly developed spraying systems that can produce an optimal size of disinfectants to reduce airborne microbes. The system was evaluated to reduce the infection caused by avian pathogenic *Escherichia coli* (APEC), an airborne bacterium which survives in indoor spaces. pH-neutral electrolyzed water (NEW) containing 100 ppm of free chlorine, laboratory-scale chambers, a recently developed sprayer, and a conventional sprayer were used in the study. A total of 123 day-of-hatch male layer chicks (Hy-Line W-36) were randomly classified into five groups (negative control (NC): no treatment; treatment 1 (Trt 1): spraying only NEW without APEC; treatment 2 (Trt 2): spraying NEW + APEC using a high-pressure sprayer; treatment 3 (Trt 3): spraying NEW + APEC using a conventional sprayer; positive control (PC): spraying only APEC). Experimental chicks in the chambers were daily exposed to 50 mL of NEW and/or APEC (1.0 × 10^6^ cfu/mL) until the end of the experiment (day 35). APEC strains were sprayed by ViKiller^®^. At least four chicks in each group were evaluated weekly to monitor APEC infection and determine the lesion. Data showed that our spraying system significantly reduced airborne APEC concentrations, mortality rate, respiratory infection, and APEC lesions in birds in the chamber space (*p* < 0.05). The results demonstrate that the antibacterial effect of the novel spraying sprayer with NEW on APEC was far superior compared to the conventional sprayer. This study provides a new insight for preventive measures against airborne microorganisms in indoor spaces.

## 1. Introduction

Pathogenic agents, including viruses, prions, bacteria, parasites, and fungi, can be transmitted and contaminated into animal houses via a number of routes such as the air, diet, water, workers, etc. [1,2,3]. Pigs and poultry are normally raised within enclosed facilities making them vulnerable to the growth and transmission of pathogenic organisms which can cause infectious disease and economic loss in farms. In particular, pigs and poultry are normally raised within enclosed facilities. Thus, indoor poultry and pig farms contain more airborne contaminants compared to outdoor cattle farms. Particularly, most of the poultry houses are built with high-rise cages, deep litter, or belts to produce white meat or eggs, and numerous bioaerosols, including viruses and bacteria. It is known that microscopic pathogens are commonly found in poultry houses and transmitted between animals via air [2,4,5]. For these reasons, the proper implementation of disinfection protocols is necessary in reducing pathogenic organisms in indoor spaces of animal buildings [6,7].

Previous studies showed that poultry house disinfection is considerably effective to increase weight gain and decrease mortality in flocks by reducing pathogenic microbes in the houses [8,9]. Methods for disinfection in poultry houses are widely used via spraying, thermo-misting, and foaming with disinfectants such as phenols, quaternary ammonium compounds, or aldehydes [10,11]. However, evaluation of the efficacy in reducing the microorganisms in the indoor spaces of poultry houses has been limited. Thus, indoor-space disinfection from bioaerosols with the surface decontamination in poultry houses is important to decrease the biological risks of contamination in poultry farming by providing sufficient time to contact airborne microbes. In addition, the World Health Organization (WHO) reported that 10 to 30 μm diameter of fog droplets are optimal size for space disinfection [12].

Avian pathogenic *Escherichia coli* (APEC) causes colibacillosis which is one of the most common infectious bacterial diseases of the layer industry. It is a syndrome caused by *E. coli* found in the gastrointestinal tract of birds and disseminated widely in feces [13]. APEC was also known to aerobically transmit between flocks as an airborne pathogen [14,15]. The infections caused by APEC can cause serious financial damage in the broiler industry worldwide by causing perihepatitis, pericarditis, omphalitis, airsacculitis, egg peritonitis, splenomegaly, etc. [13]. Previous studies showed that numerous airborne bacteria, including APEC, are present within poultry houses [16,17]. Accordingly, APEC causes colibacillosis which also causes increased mortality and morbidity leading to financial losses in the poultry industry. Therefore, reducing the growth of APEC in poultry houses is highly needed to alleviate its harmful effects, which include the transmission of colibacillosis, and consequently increase poultry farm income via an improvement in meat and eggs production.

Currently, poultry farm disinfection is commonly performed during the period between production cycles through foaming, spraying, or fogging systems with various commercially available disinfectants. However, conventional disinfection strategies are not sufficient methods for proper indoor disinfection specifically during production cycles due to the high toxicity levels of disinfectants, and the equipment is limited in producing the optimum particle size of the disinfectants for an extended period of time to sufficiently contact airborne microbes. Accordingly, we have developed a novel spraying system that consists of a high-pressure (HP) sprayer and pH-neutral electrolyzed water (NEW, main component—hypochlorous acid (HOCl)) disinfectant to effectively reduce airborne microbes in indoor spaces through the production of an optimal size (10~30 μm) of the disinfectant for ensuring efficacy and safety in any given time and space. Therefore, the aim of the present study was to evaluate our HP-spraying equipment and disinfectant to prevent the transmission of harmful APEC between experimental chicks grown in indoor spaces.

## 2. Materials and Methods

### 2.1. Cultures, Experimental Animals, and Diet

Avian pathogenic *E. coli* (APEC) provided by the Avian Disease Laboratory (ADL) of Konkuk University was grown in Luria–Bertani (LB, Difco Laboratories, Detroit, MI, USA) broth at 37 °C for 18 h and stored in LB containing 20% glycerol at −80 °C until use. One-hundred twenty-three Hy-Line Brown male layers used in this study were kindly provided by a commercial hatchery (Korean Poultry TS Co., Ltd., Icheon-si, Gyeonggi-do, Korea). They were fed with grower feed purchased from AT Immune Inc. (Cheongju, Choongbuk, Korea) until the end of the experiment period (day 35). There was no supplementation of feed with either antibiotic agents or growth promoters during the experimental period.

### 2.2. Experimental Chambers and Spraying System

The experiments for airborne bacterial reduction and antimicrobial activity against APEC strains in layers using the HP sprayer with pH-neutral electrolyzed water (NEW, pH 7.0) disinfectant were conducted in the laboratory-scale chambers (3.0 m × 1.5 m × 2.0 m and 2.0 m × 1.0 m × 2.0 m) as shown in Figure 1a,b, respectively. This NEW (Deger100^®^, LGS Corporation, Gwangmyeong-si, Korea) was produced from salt water. The spraying system (ViKiller^®^, TracoWorld Ltd., Gwangmyeong-si, Korea) that mainly consisted of an HP pump (UHP S1, TracoWorld Ltd.), nozzles (LYOHM^®^, H. Ikeuchi & Co., Ltd., OSAKA, Japan), battery (G-MAX^®^, GreenWorks, Mooresville, CA, USA), and NEW disinfectant was developed to reduce airborne microbes in an indoor space. A conventional sprayer (DH-Fog30, DeahoGreen, Kimhae-si, Korea) with a low-pressure venturi nozzle was also used to compare to our HP sprayer. The distribution of spray particles’ sizes produced by the HP sprayer was measured using the Spraytec (Malvern Panalytical Ltd., Malvern, UK).

### 2.3. Experimental Design

A preliminary study for determining the effective concentrations of NEW against APEC was performed to further evaluate the antimicrobial activity on airborne APEC in the space for layers. The concentrations of APEC tested in the study ranged from 1.0 × 10^5^ cfu/mL to 1.0 × 10^9^ cfu/mL. The NEW used in the study contained mainly HOCl with 100 to 500 ppm free chlorine. For space disinfection experiments to determine the antimicrobial activity of NEW against APEC, fifty milliliters of APEC cultures diluted with PBS was sprayed into the chambers with or without the same amount of NEW. In each group, the APEC in the space of the chambers fell on three TSA plates and the lids of the plates were opened and closed at 0 and 3, 3 and 6, 6 and 9, and 9 and 12 min after spraying APEC followed by NEW in the chamber (Figure 1a). The plates were incubated at 37 °C for 16 h and cell colonies were then counted. For the clinical experiment, a total of 123 day-of-hatch male layer chicks (Hy-Line W-36) were randomly classified into 5 groups (negative control (NC): no treatment; treatment 1 (Trt 1): spraying only NEW without APEC; treatment 2 (Trt 2): spraying NEW + APEC using a HP sprayer; treatment 3 (Trt 3): spraying NEW + APEC using a conventional sprayer; positive control (PC): spraying only APEC). APEC and NEW were sprayed using the HP sprayer and the HP sprayer for Trt 2 or a conventional (ULV) sprayer for Trt 3, respectively. Experimental chicks in the chambers were daily exposed to 50 mL of NEW (100 ppm) and/or APEC (1.0 × 10^6^ cfu/mL) at 4:00 p.m. until the end of the experiment (day 35). A diet was daily provided at 9:00 a.m. and 5:00 p.m. Feed and drinking water were provided ad libitum through the whole experimental period. The temperature in the chambers was set to 30 °C until the end of the experiment. The chicks were illuminated with a 23 h light and 1 h dark–light schedule throughout a week’s experimental period. The half hour for dark time was then increased weekly until day 35 (Figure 2). Manure and litter on the bottom of the chambers were weekly removed and replaced, respectively. At least 4 chicks in each group were selected randomly and sacrificed weekly to evaluate APEC infection. The experiment was conducted from April to June of 2021. All animal and experimental procedures were approved by the Institutional Animal Care and Use Committee at Konkuk University (approval number-KU21025).

### 2.4. Mortality and Lesion

Four layers in the chambers from each group were weekly transferred to the analytical laboratory from the KU poultry house to observe classic lesions of colibacillosis, including airsacculitis, pericarditis, perihepatitis, splenomegaly, and osteoarthritis. Dead layers were daily recorded.

### 2.5. Statistical Analysis

Data for viable cell numbers of the APEC strain and mortality rates were analyzed by one-way ANOVA followed by Tukey HSD post hoc test using Excel 2019 (Microsoft, Redmond, WA, USA). The statistical significance of differences among the group means was considered at a *p*-value of less than 0.05.

## 3. Results

### 3.1. Distribution of the Droplet Size of NEW by a Novel Spraying System

Data obtained from the spray particle size analyzer (Malvern Panalytical Ltd.) presented that over 90% the particle sizes of NEW generated by the ViKiller^®^ (TracoWorld Ltd.) ranged from 10 to 30 μm as shown in Figure 3. The constant and ultra-fine-sized NEW droplets were produced by an HP pump and an optimal size of spray nozzle (specifications not shown). The disinfectant particles were shown floating for 5 min in the experimental chambers via observing the cloudiness inside the space of the chambers.

### 3.2. Determination of the Effective Concentrations of NEW against APEC

The effective concentrations of NEW on the anti-survival of APEC strains in the air were measured, and the recovered cell numbers from each group are shown in Table 1. The number of cells recovered for 3 min after spraying APEC followed by NEW was perceptibly decreased up to 12 min according to increased NEW concentrations. In addition, bacterial aerosols were shown to be decreased with increasing amount of time (Table 1). The bioaerosols were consistently floated in the space of the experimental chambers up to 12 min after spraying. No APEC strain was found by using over 100 ppm NEW from 3 min after spraying a low concentration of APEC (1.0 × 10^6^ cfu/mL). Figure 4 also shows the antibacterial activity of NEW against the APEC strain. The viable cell numbers of APEC sprayed into the chambers with or without NEW (Deger^®^ containing 100 and 200 ppm of free chlorine) were decreased in a time-dependent manner (Figure 4a). The APEC in the air was nearly reduced from 3 min after spraying 100 and 200 ppm NEW. Based on the data, we used 1.0 × 10^6^ cfu/mL APEC and 100 ppm NEW for further clinical study. Figure 4b presents the viable cells recovered from groups treated with and without NEW. The data show a 99.9% reduction in APEC by spraying NEW in the experimental chambers (Figure 4b).

### 3.3. Observation of Lesion and Determination of Mortality Rate in Clinical Study

At least four layers weekly selected from each group were euthanized by carbon dioxide (CO2) gas and sacrificed to observe lesions of colibacillosis. Figure 5a shows APEC-afflicted layers with gross lesions of airsacculitis, pericarditis, perihepatitis, splenomegaly, and osteoarthritis. Most of the layers in Trt 3 and PC groups had many more gross lesions than Trt 1 and 2 (Figure 5b) as well as the NC group (Table 2). The dead layers in the Trt 3 and PC groups were also more than those in NC, Trt 1, and Trt 2. (Table 2). The layers in the Trt 1 group treated with only NEW were shown to be mostly as normal as the NC group, whereas the PC and Trt 3 groups had many chicks with severe lesions (Table 2). In addition, the numbers of normal chicks and severely APEC-afflicted chicks in Trt 2 were shown to be much higher and lower, respectively, than that in Trt 3. The novel spraying system using the HP sprayer, producing a 10~30 μm droplet size, with NEW showed a higher efficacy to reduce APEC strains in the indoor space compared to the conventional sprayer, which displayed more pathologically normal layers, less severe lesions, and low mortality rates (Figure 6).

## 4. Discussion

Poultry-farming environments currently contain numerous biological risks for various contaminants with avian infectious viruses, bacteria, fungi, or parasites [2,4]. Based on our recent data, 61 bacterial species, including pathogenic *E. coli*, *Enterobacter cloacae*, *Klebsiella pneumoniae*, *Pantoea agglomerans*, *Proteus mirabilis*, *Salmonella* sp., and *Staphylococcus aureus*, were isolated and identified from layers’ swabs, diet, water, floor, feces, fans, and workers’ gloves in a poultry house (unpublished data). Thus, a fogging, misting, or foaming system for disinfecting the environments including the floor, cages, belt, deep litter, etc., in poultry houses has been introduced and used as a measure to control infectious diseases [10,11]. Ozone, chlorine, quaternary ammonium salt, and glutaraldehyde compounds are commonly used in poultry houses [10]. As far as we know, there has been no study conducted related to the inactivation of airborne APEC in indoor spaces utilizing the HP sprayer and NEW through space disinfection. Additionally, there is no disinfectant approved in any country or the EU for inactivating airborne microorganisms during the animal production cycle, and only surface disinfectants to remove pathogens are approved and commercially available to date. Due to the lack of knowledge and technology to effectively reduce airborne pathogens, we have developed a spraying system that can produce an effective size of disinfectant droplets to inactivate the airborne pathogens via extending floating time to contact them through HP spraying. Consequently, space disinfection is becoming a vital step as a crucial measure to the health security and safety of both humans and animals exposed to airborne pathogens.

ViKiller^®^ and NEW as the HP sprayer and disinfectant, respectively, were used to generate an optimal droplet size (10~30 μm) and reduce APEC in an indoor space. NEW demonstrates several advantages since it is natural, nontoxic, nonselective, nonirritant, and noncorrosive. First, NEW has been known as an ecologically and environmentally safe disinfectant, which does not contain a toxic contaminant [18]. HOCl is produced by the heme protein myeloperoxidase using hydrogen peroxide (H_2_O_2_) and chloride in the granulocytic cells of our body [19]. This NEW is produced from salt water and returned to normal water after use [20]. Second, NEW has an ability to inactivate bacterial pathogens and nonselective property to antimicrobials or disinfectants [21,22,23]. According to a previous study, over >6.0 log10 *E. coli* contaminated in eggs experimentally was reduced by treatment of NEW for 30 s [24]. We also observed the efficacy of NEW against other pathogens, including *Staphylococcus aureus* and *Pseudomonas fulva* (a dominant airborne bacterium in animal and human buildings) via a reduction > 4.0 log10 (unpublished data). Third, NEW is a neutral, safe, and stable disinfectant with high antimicrobial activity (compared to chlorine bleach) in storage and transport. Thus, it can be used in any space and at any time. In addition, NEW does not negatively damage human skin and the mucous membrane [3,25].

The utilization of HOCl water for antimicrobial activity was initially introduced in 1987 and its application has been considerably increased, especially in the fields of agriculture, food processing, dentistry, ophthalmology, dermatology, and medicine [3,6,9,18,24,25]. HOCl is endogenously produced and effective against a wide range of microorganisms [6,26]. Almost all disinfectants, including HOCl, have been used for surface disinfection up until now. Previous studies showed the significant antimicrobial activity of HOCl (40~300 ppm) against airborne microorganisms in animal houses [6,7,27,28]. Hao and colleagues showed that 40 ppm HOCl was effective in inactivating airborne *Salmonella, S. aureus*, and Coliforms [6]. However, fungi were completely inactivated by spraying 300 ppm HOCl. Tamaki et al., demonstrated that NEW with ≥40.4 ppm free available chlorine reduced 3.0 log H5N1 and 5.5 log H9N2 viruses in aqueous suspension [28]. Park et al. revealed the effects of a liquid or fog form HOCl on the reduction in human norovirus on the surfaces of ceramic tile and stainless steel [27]. Additionally, Zhao and colleagues presented that membrane-less acidic electrolyzed water (MLAEW, pH 6.0~7.0) aerosols by spraying significantly decreased airborne bacteria in a layer house [7]. In the current study, Deger^®^ containing HOCl completely inactivated airborne APEC in the experimental chambers. Therefore, the HOCl-spraying method, providing a sufficient time to contact airborne microorganisms, can be applied to effectively inactivate airborne bacterial and viral pathogens for space disinfection of animal houses as well as surface disinfection.

Recently, the US Environmental Protection Agency (EPA) has approved the use of HOCl as a disinfectant agent against COVID-19. Over 30 HOCl products for surface and food disinfection were registered on the list N to kill the coronavirus SARS-CoV-2 (COVID-19) when used according to directions [29]. According to a previous study, seven mice had free access to acid electrolyzed functional water (AEFW, pH ≤ 2.7, 20~60 ppm free chlorine) as drinking water and then their systemic and gastrointestinal changes from AEFW ingestion were evaluated [30]. They found no histopathological and morphological changes in gastrointestinal tracts, teeth, and tooth enamel surface as well as body weight between control (tap water, pH 7.5, 0.5 ppm free chlorine) and test groups after 8 weeks. In addition, the cleaning and disinfecting (C&D) method by misting HOCl (1000 ppm) reduced ATP scores in high-touch areas of ambulatory surgery center rooms significantly more than a standard C&D method using quaternary ammonium and benzalkonium chloride that can be toxic to staff [31]. For space disinfection utilizing NEW as a safe disinfectant, further studies are required to ensure the safety against future risks such as acute and chronic toxicity, carcinogenicity, reprotoxicity, and genotoxicity through inhalation. NEW, mainly containing HOCl, also did not negatively affect the health of the experimental chicks during the whole experiment period (35 days) in the current study. In conclusion, the novel high-pressure spraying system with NEW can effectively inactivate airborne bacteria in poultry houses by providing sufficient time of contact between pathogen and disinfectant.

## Figures and Tables

**Figure 1 microorganisms-10-02201-f001:**
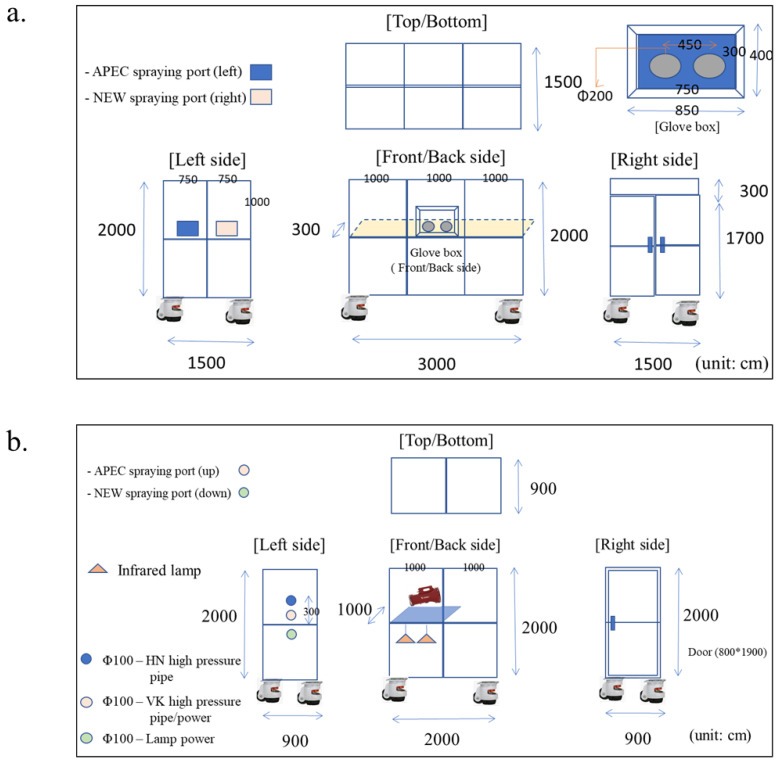
The designs of laboratory-scale chambers. Chamber (3.0 m × 1.5 m × 2.0 m) was used to determine the effective concentrations of NEW on the reduction in APEC strains in the space (**a**). Chamber (2.0 m × 1.0 m × 2.0 m) was used to measure the antimicrobial activity of NEW against APEC (**b**).

**Figure 2 microorganisms-10-02201-f002:**
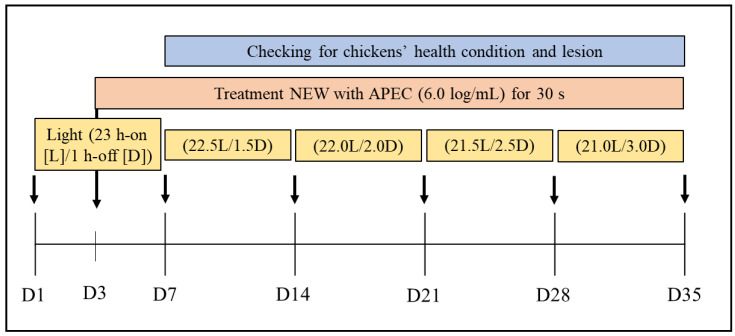
Experimental design and schedule for the antimicrobial activity of NEW against APEC. Male layer chickens in treatment groups were daily exposed to 50 mL of APEC strains (1.0 × 10^6^ cfu/mL), followed by 50 mL of NEW (100 ppm free chlorine) with a novel high-pressure sprayer or a conventional sprayer after day 3. The experimental chicks were illuminated with a 23 h light and 1 h dark–light schedule throughout a week’s experimental period. Half hour for dark time was then increased weekly until day 35. At least four layers in the chambers from each group were sacrificed weekly to evaluate APEC infection via observation of colibacillosis lesions.

**Figure 3 microorganisms-10-02201-f003:**
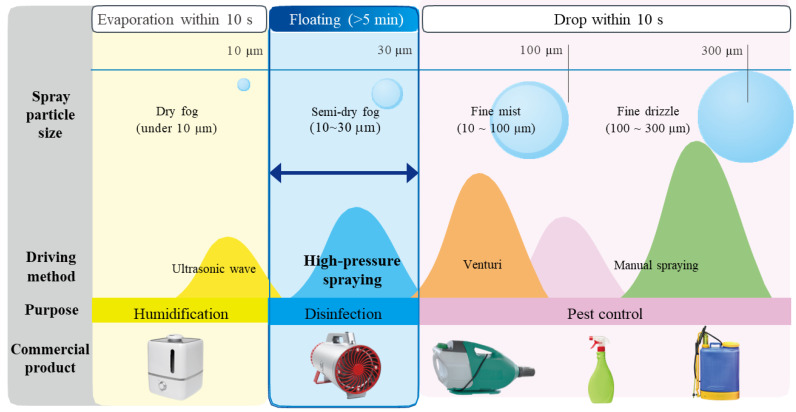
Size distribution of droplets generated by various sprayers. Various commercial products used in humidification, disinfection, and pest control were used to compare the particle sizes of spray at different stages at specific times. Various products apply specific driving methods such as ultrasonic wave, high-pressure spraying, and venturi/manual spraying utilized for purposes such as humidification, disinfection, and pest control, respectively. The machine analyzer from Malvern Panalytical Ltd. was utilized to determine the specific spray particle size. Sizes of particles were measured in 3 stages: Stage 1: evaporation within 10 s; Stage 2: floating for about 5 s; Stage 3: drop within 10 s. Particle sizes are classified into four, namely, dry fog (<10 μm), semi-dry fog (10–30 μm), fine mist (10–100 μm), and fine drizzle (100–300 μm). Results showed that over 90% of the detected particle sizes ranged from 10 to 30 μm which is the effective range of spray particle size when NEW generated by the ViKiller^®^ (TracoWorld Ltd.) was used for disinfection.

**Figure 4 microorganisms-10-02201-f004:**
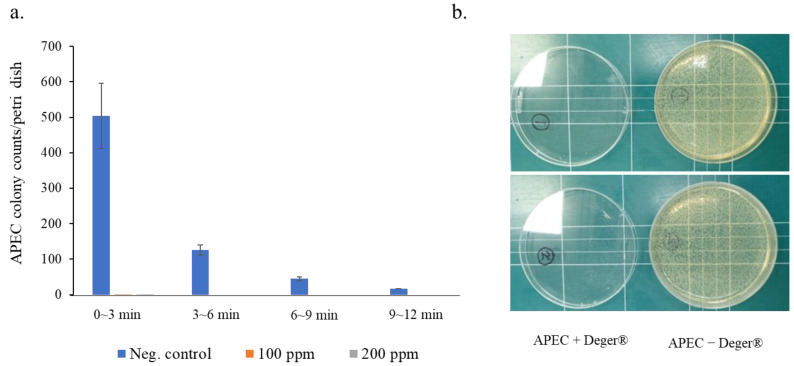
Effective concentrations of NEW on the anti-survival of APEC strains in the air. The number of cells was determined for 3 min after spraying APEC followed by NEW up to 12 min. APEC strains in the chambers were perceptibly decreased up to 12 min according to increased NEW concentrations (**a**). The viable cells were recovered from groups treated with and without NEW. Fifty mL of APEC strains (1.0 × 10^6^ cfu/mL) in the experimental chambers was considerably reduced from 3 min after spraying 50 mL of 100 ppm NEW. The pictures show the 99.9% reduction in APEC by spraying NEW (**b**).

**Figure 5 microorganisms-10-02201-f005:**
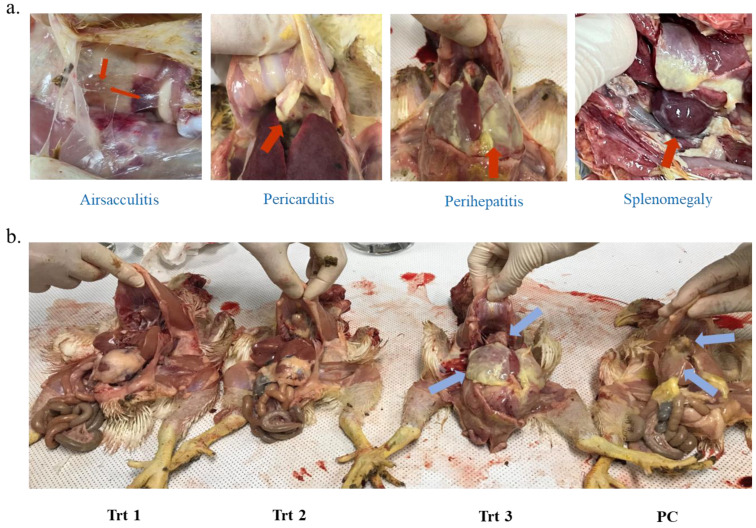
Lesions in the experimental chicken. Red arrows indicate that gross lesions from APEC-afflicted layers with airsacculitis, pericarditis, perihepatitis, and splenomegaly are presented (**a**). Blue arrows show that severe pericarditis and perihepatitis occurred in the Trt 3 and PC groups, whereas there were no symptoms in the Trt 1 and Trt 2 groups (**b**).

**Figure 6 microorganisms-10-02201-f006:**
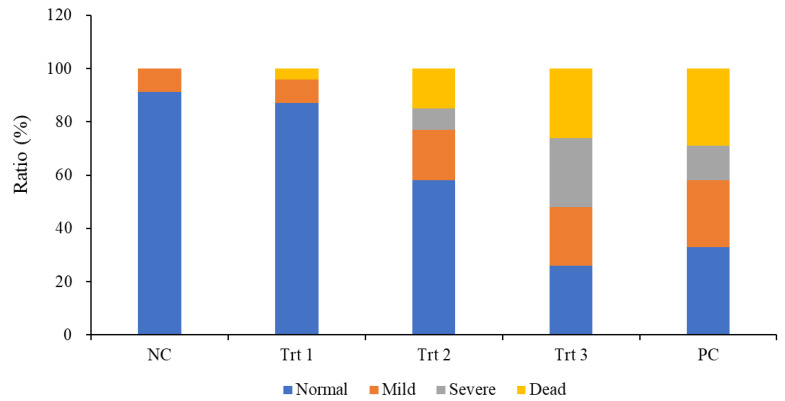
Symptoms and mortality rates in the experimental chicken. More severe symptoms were found in Trt 3 and PC groups. APEC also caused relatively higher death rates in the groups. Death rates and severe symptoms in Trt 2 were much lower than in Trt 3 and PC groups.

**Table 1 microorganisms-10-02201-t001:** APEC strain recovered from each group used in the study.

Cell Collection Time after Spraying APEC Followed by NEW	0~3 min		3~6 min		6~9 min		9~12 min	
Cell Numbers Sprayed (cfu/mL)	1.0 × 10^6^	1.0 × 10^9^	1.0 × 10^6^	1.0 × 10^9^	1.0 × 10^6^	1.0 × 10^9^	1.0 × 10^6^	1.0 × 10^9^
Neg. control	504	>10^3^	126	>10^3^	45	242	15	141
100 ppm	3	49	0	27	0	34	0	27
200 ppm	1	17	0	13	0	7	0	51
500 ppm	0	2	0	2	0	1	0	2

**Table 2 microorganisms-10-02201-t002:** Data for the lesions and mortality rates by APEC in experimental layers.

Degree of Lesion	Normal	Mild	Severe	Dead	Total
Group					
Negative control (no APEC + NEW)	21	2	0	0	23
Trt 1 (only NEW)	20	2	0	1	23
Trt 2 (APEC + NEW using ViKiller^®^)	15	5	2	4	26
Trt 3 (APEC + NEW using ULV sprayer)	7	6	7	7	27
Positive control (only APEC)	8	6	3	7	24

## Data Availability

Not applicable.

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
