# Peer review of "Inactivation of Airborne Avian Pathogenic E. coli (APEC) via Application of a Novel High-Pressure Spraying System"

_microorganisms, 2022, doi:10.3390/microorganisms10112201_

Round 1

Reviewer 1 Report

The authors developed a novel spaying system comprising a high-pressure sprayer and hypochlorous to produce an optimal size of disinfection for effectively removing airborne microbes. The goodness of the newly developed device was evaluated according to the bacterial growth (colony formation) in a well-controlled laboratory condition and the survivability of the bacterial-affected chickens. The results showed that the device was applicable. 

I'm afraid I couldn't find the scientific question addressed by the present study. I feel that the manuscript is more like a technical report of the newly developed device rather than a scientific paper of new findings because it neither provides any scientific understanding nor solves any scientific issues. There's no result showing the spray systems produced the particle of the size they supposed and how such a sized particle could inhibit bacterial growth. There's no explanation why the experimental design was reasonable and reliable. All the results demonstrated that the newly developed spray system was useful. The authors may think about what the new scientific finding of the present study is. 

Author Response

Your points are greatly appreciated. We believe that our study is both a technical and scientific report related to the advantages of creating a newly developed device which we believe is helpful to provide necessary information and helpful solution to current scientific issues especially dealing with airborne pathogens which are constantly evolving, growing, and increasing with the advent of pandemic and global climate change issues. We are the first in the world who have developed a portable high-pressure pump and nozzle to create effective particle size for efficient air disinfection. Our newly developed technology allows 10~30 mm particle size of disinfectant for long-lasting suspension in space for effective disinfection which is in accordance with the guidelines of World Health Organization. In this study, we compared the effect of different particle sizes produced by a conventional sprayer (ULV) and our high-pressure sprayer against inactivation of APEC. However, we did not open the entire technical data due to protection of intellectual properties and infringement of our company’s interests. Consequently, the current study shows limited data that the technological advancement and sterilization performance of the developed equipment are consistent with the intended clinical evaluation. We appreciate your accurate point again.

Reviewer 2 Report

Dear Author,

This is a well designed study with merit since it evaluates the efficacy of novel high-pressure spraying system in inactivation of airborne APEC.

This is a well-written manuscript. However, it needs some changes in the conclusions. 

Page 10, Line 308-312: Do not include general statements in the conclusions like " HOCl denatures bacterial membrane or viral capsid proteins" or " NEW containing the oxidizing agent can effectively inactivate airborne bacteria or virus". Instead, include study-specific conclusions which include the effect of NEW on APEC. Since, you have not tested the efficacy of NEW on any virus, do not include that in the conclusions. 

Author Response

We appreciate your suggestion and comment. We revised and modified our manuscript as you indicated.

(Revised Manuscript L308-312) As indicated we have modified the statement “In conclusion, the novel high-pressure spraying system with NEW can effectively inactivate airborne bacteria in poultry houses by providing sufficient time of contact between pathogen and disinfectant.”